# A Learning-based Iterative Method for Solving Vehicle Routing Problems

**Hao Lu** *
Princeton University, Princeton, NJ 08540
{haolu}@princeton.edu

**Xingwen Zhang** * **& Shuang Yang**
Ant Financial Services Group, San Mateo, CA 94402
{xingwen.zhang,shuang.yang}@antfin.com

## Abstract

This paper is concerned with solving combinatorial optimization problems, in particular, the capacitated vehicle routing problems (CVRP). Classical Operations Research (OR) algorithms such as LKH3 (Helsgaun, 2017) are inefficient and difficult to scale to larger-size problems. Machine learning based approaches have recently shown to be promising, partly because of their efficiency (once trained, they can perform solving within minutes or even seconds). However, there is still a considerable gap between the quality of a machine learned solution and what OR methods can offer (e.g., on CVRP-100, the best result of learned solutions is between 16.10-16.80, significantly worse than LKH3's 15.65). In this paper, we present "Learn to Improve" (L2I), the first learning based approach for CVRP that is efficient in solving speed and at the same time outperforms OR methods. Starting with a random initial solution, L2I learns to iteratively refine the solution with an improvement operator, selected by a reinforcement learning based controller. The improvement operator is selected from a pool of powerful operators that are customized for routing problems. By combining the strengths of the two worlds, our approach achieves the new state-of-the-art results on CVRP, e.g., an average cost of 15.57 on CVRP-100.

## 1 Introduction

In this paper, we focus on an important class of combinatorial optimization, vehicle routing problems (VRP), which have a wide range of applications in logistics. Capacitated vehicle routing problem (CVRP) is a basic variant of VRP, aiming to find a set of routes with minimal cost to fulfill the demands of a set of customers without violating vehicle capacity constraints. The CVRP is NP-hard (Dantzig & Ramser, 1959), and both exact and heuristic methods have been developed to solve it (Fukasawa et al., 2006; Golden et al., 2008; Kumar & Panneerselvam, 2012; Toth & Vigo, 2014).

In recent years, especially after the seminal work of Pointer Networks (Vinyals et al., 2015), researchers start to develop new deep learning and reinforcement learning (RL) frameworks to solve combinatorial optimization problems (Bello et al., 2016; Mao et al., 2016; Khalil et al., 2017; Bengio et al., 2018; Kool et al., 2019; Chen & Tian, 2019). For the CVRP itself, a number of RL-based methods have been proposed in the literature (Nazari et al., 2018; Kool et al., 2019; Chen & Tian, 2019). The learning based methods are trained on a huge number of problem instances, and have been shown to be extremely fast in producing solutions of reasonably good quality. However, when tested with the same benchmark instances, these learning-based methods cannot outperform the state-of-the-art method LKH3 (Helsgaun, 2017), which is a penalty-function-based extension of classical Lin-Kernighan heuristic (Lin & Kernighan, 1973; Helsgaun, 2000). For example, on CVRP with 100 customers, LKH3 is able to produce an average cost of 15.65. This line of research motivated us to study a framework that combines the strength of Operations Research (OR) heuristics with learning capabilities of machine learning (RL in particular). Machine learning can learn

---

*Equal contribution.

to solve a class of problem instances fast, when test instances are generated from the same distribution as training instances. Classical approaches like search algorithms are effective but may need heavy computation, which is time-consuming. Our research interest is in fusing the strengths of these two worlds. Another related line of research is hyper-heuristics, which is "a search method or learning mechanism for selecting or generating heuristics to solve computational search problems" (Burke et al., 2013). Instead of developing a high-level methodology without the need of knowing the details of low-level heuristics, we are primarily interested in a closely integrated system that best utilizes the strength of OR operators and learning capability.

**Our Contributions**. Instead of directly constructing a solution from the problem instance (Graves et al., 2014; Sutskever et al., 2014; Vinyals et al., 2015), we propose a framework that iteratively searches among solutions, until a certain termination condition is satisfied. Our main contributions are as follows:

- We present a learning-based algorithm for solving CVRP, achieving new state-of-the-art results. The recent line of works using RL to solve CVRP shows the potential of machine learning algorithms. They can solve CVRP faster, but cannot beat classical OR solvers like LKH3 in term of solution quality. Our algorithm is the first machine learning framework that outperforms LKH3 on CVRP.

- We propose a novel hierarchical framework. Instead of putting all operators in one action pool, we separate heuristic operators into two classes, namely improvement operators and perturbation operators. At each state, we choose the class first and then choose operators within the class. Learning from the current solution is made easier by focusing RL on the improvement operators only.

- We propose an ensemble method, which trains several RL policies at the same time, but with different state input features. The ensemble method is shown to produce superior results than individual policies with an equivalent amount of computation.

**Related Work**. In recent years, there have been many studies using deep learning and RL to solve combinatorial optimization problems (Smith, 1999; Mao et al., 2016; Lodi & Zarpellon, 2017; Veličković et al., 2017; Lombardi & Milano, 2018; Bengio et al., 2018). Routing problems, especially traveling salesman problems (TSP) and VRP, have been explored by a sequence of works (Vinyals et al., 2015; Bello et al., 2016; Khalil et al., 2017; Li et al., 2018; Deudon et al., 2018; Kaempfer & Wolf, 2018; Nazari et al., 2018; Kool et al., 2019; Chen & Tian, 2019). Most of these works, with the exception of Chen & Tian (2019), follow an end-to-end approach, which is directly constructing a solution from scratch. Vinyals et al. (2015) first introduce the Pointer Network, inspired by sequence-to-sequence models, to solve TSP. They use an attention model to learn the order of different nodes in a supervised fashion. Later Bello et al. (2016) develop an RL algorithm to train the Pointer Network. Their framework learns the optimal policy from problem instances and needs no supervised solutions. Nazari et al. (2018) improve the Pointer Network with a new design, making the model invariant with respect to the input sequence, and extend it to solve VRP. Kool et al. (2019) propose a model based on attention layers, and an RL algorithm to train this model with a simple but effective baseline. Chen & Tian (2019) propose a NeuRewriter model for VRP. They define a rewriting rule set and train two policy networks, a region-picking policy and a rule-picking policy, to obtain the next state. Given an initial solution, their goal is to find a sequence of steps towards the solution with minimal cost.

## 1.1 NOTATIONS

In CVRP, there is a depot and a set of $N$ customers. Each customer $i$, $i \in \{1, \ldots, N\}$, has a demand $d_i$ to be satisfied. A vehicle, which always starts at and ends at the depot, can serve a set of customers as long as the total customer demand does not exceed the capacity of the vehicle. The traveling cost $c_{i,j}$ is the cost of a vehicle going from node $i$ to $j$, with $i, j \in \{0, 1, \ldots, N\}$ (where the depot is denoted by node 0 for convenience). The objective is to find a routing plan with minimal cost that serves all customers without violating vehicle capacity constraints. Figure 1 gives an illustration of CVRP, while a mathematical formulation is given in the Appendix.

**Route**. A route is a sequence of nodes visited in order, with the depot (node 0) as the starting and ending node. For example, [0, 1, 3, 4, 0] is a traveling plan that starts at the depot, visits node 1, 3 and 4 sequentially, and returns to the depot.

**Solution**. A solution is a set of routes such that each customer is visited exactly once, and the total demand along each route is less than the vehicle capacity.

**Operator**. An operator is a mapping from one solution to another. In this paper, instead of directly constructing a solution from scratch, we improve or perturb the solution iteratively using operators.

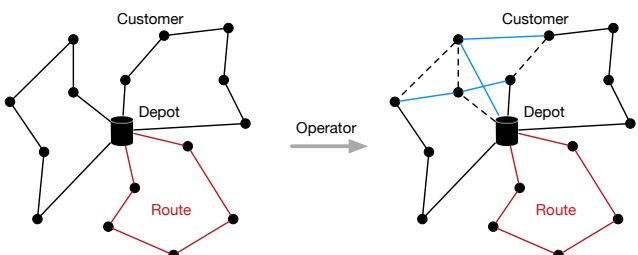

Figure 1: An illustration of CVRP. Here we provide a problem instance. The red one is a sample route, and three routes consist of a solution for this problem instance. After applying an operator, current solution changes to a new solution with dashed lines replaced by blue lines.

## 2 LEARN TO IMPROVE

In this section we formally introduce our "Learn to Improve" (L2I) framework, including main components of the system as well as the design intuitions behind individual components. Figure 2 illustrates the overall framework. As mentioned previously, the framework is iterative in nature, that is, we always start with a feasible solution, continuously improving the solution or perturbing it. Along the way, all constraints remain satisfied. By always maintaining the feasibility of the solution, we are exploring the space of feasible solutions and any of them found by our search process could potentially be a good solution. After $T$ (a parameter set in advance) steps the algorithm stops, and we choose the one with the minimum traveling cost as our final solution. Our framework has a few distinct components (e.g., how to improve the solution, when and how to perturb it), which could be rule-based, learned by machine learning or mixed, and thus allows us to experiment with different ways of integrating machine learning and OR, some of which could lead to superior methods either in terms of better solution quality or in terms of computational efficiency. It is worthwhile to point out that, as contrast to the work of Chen & Tian (2019), the solution space we explore is much larger since we are employing a rich set of improvement and perturbation operators.

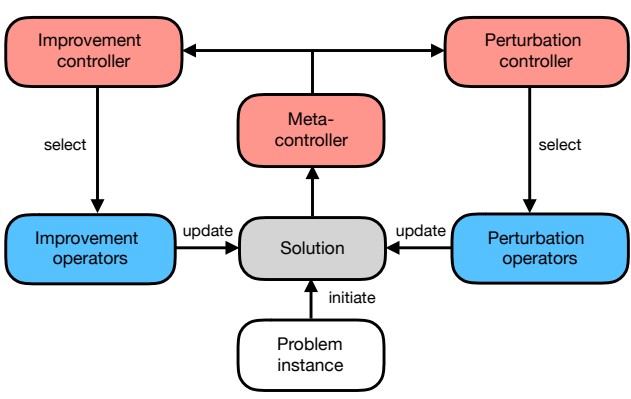

Figure 2: Our hierarchy framework. Given a problem instance, our algorithm first generates a feasible solution. Then it iteratively updates the solution with an improvement operator selected by an RL-based controller or with a perturbation operator chosen by a rule-based controller. After a certain number of steps, we choose the best one among all visited solutions.

For this research work, we implemented and experimented with a number of design choices, and ended up with a method that is both computationally efficient and able to produce state-of-the-art results. The details of the main components of our method are as follows. Given a history of most recent solutions, our method uses a threshold-based rule to decide whether we should continue to improve the current solution, or should perturb it and restart with the perturbed solution. If it decides

that the current solution could still be improved, it will use an RL-based controller to choose one of the improvement operators, and try to improve the solution with the chosen operator. We have a rich set of improvement operators (the list of improvement operators and their details are given in the Appendix), where intra-route ones attempt to reduce the cost of current solution by moving customers to different positions in individual routes, while inter-route ones attempt to reduce the cost by moving customers among different routes. Given that the improvement operators are of distinct characteristics, it is not straightforward to know in advance which operators are most effective for the problem under investigation. It is also difficult to know a pre-defined ordering of the operators that is best for the problem. Thus, an RL-based controller is a good choice to learn the set of improvement operators that is more effective for the problem (or more specifically, the problem and the data distribution from which the training and test instances are sampled). As will be shown in Section 3, our RL model is able to differentiate more useful improvement operators from less useful ones for CVRP, as well as generate an implicit ordering on how the operators will be applied.

On the other hand, upon reaching a local minimum the perturbation controller randomly chooses a perturbation operator that destroys (completely or partially) and reconstructs a number of routes to generate a new starting solution (Table 6 in the Appendix gives the list of perturbation operators and their details). Specifically, if no cost reduction has been made for $L$ improvement steps, we perturb the solution and restart the improvement iteration (where for ease of explanation a maximum consecutive sequence of improvement operators applied before perturbation is called an improvement iteration). As a perturbation changes the solution quite dramatically (by producing a substantially different solution that is usually worse than the current one), we found that it is useful to start a new improvement iteration with a reasonably good starting point (e.g. by filtering out the restarting solutions that are significantly worse than current solution or currently best solution). It is clear that we purposely separate improvement operators from perturbation ones, and an alternative design would be to mix them all together and have a single controller deciding which operator to apply next. However, the improvement operators are of different nature from the perturbation ones, and their impacts are different since the perturbation operators have long-lasting effect by affecting an entire improvement iteration. Our experience also suggests that learning is made easier by focusing RL on the improvement operators only. Lastly, it is worthwhile to point out that although the rule-based perturbation controller is shown to be effective, we do not rule out the possibility that it can also be learning-based.

The framework described above provides a way of combining the strength of OR operators, which are powerful since they are custom-made for routing problems, with learning capabilities of RL, which is flexible and can be adapted to a given problem and its associated data distribution. Having described our overall framework, we are now ready to present the details of the improvement controller and operators.

## 2.1 IMPROVEMENT CONTROLLER AND OPERATORS

The improvement controller starts with an initial solution, which is either constructed randomly (for the first improvement iteration)[1] or produced by a perturbation operator (for subsequent iterations), and then tries to improve it, i.e., reducing the total traveling distance without violating any constraints, by selectively applying an improvement operator in Table 5. For the RL model, the set of improvement operators constitute our action space. These operators change the solution locally and most are computationally light. With the current state as input, a neural network produces a vector of action probabilities, and the weights of the network are trained with policy gradient. Figure 6 illustrates the components of our RL model, and their details are given as follows.

### 2.1.1 STATES

Each state includes features from the problem instance, the solution and the running history. Stationary features, such as the location and demand of each customer, are considered as problem-specific since they are invariant across solutions. Solution-specific features are based on the current traveling plan. For example, given a solution, for each customer we compute its neighboring nodes that are visited before and afterwards, as well as the relevant distances. Following Ödling (2018), the

---

[1] Other construction heuristics, such as the Clarke-Wright algorithm (Clarke & Wright, 1964), can also be used.

running history includes the actions that are recently taken as well as their effects. For example, $a_{t-h}$, $1 \leq h \leq H$, is the action taken $h$ steps before current step $t$, and its effect $e_{t-h}$ is +1 if the action led to a reduction of total distance, -1 otherwise. A complete description of state features is given in Table 4, where $i^-$ and $i^+$ denote the node visited before and after node $i$, $1 \leq i \leq N$, in the solution, respectively.

### 2.1.2 ACTIONS

The actions can be classified into two classes, intra-route operators and inter-route operators. An intra-route operator attempts to reduce traveling distance of an individual route, while inter-route operators aim at reducing total traveling distance by moving customers between more than one route. The details of the operators are given in Table 5. It is worthwhile to point out that the same operator with different parameters are considered as different actions. For example, Symmetric-exchange(2) with $m = 1, 2, 3$ are considered as three distinct actions. For a given problem (or even solution), the same operator with different parameters may perform differently and thus it is appropriate to treat them as separate actions and let the RL model learn how to best use them.

### 2.1.3 POLICY NETWORK

We use the well-known REINFORCE algorithm (Williams, 1992) to update the gradient

$$\nabla_\theta J(\theta|s) = \mathbb{E}_{\pi \sim p_\theta(.|s)} \big[ (L(\pi|s) - b(s)) \nabla_\theta \log p_\theta(\pi|s) \big]$$

of policy with a baseline function $b(s)$. Given a state as described in Section 2.1.1, a policy network outputs a list of action probabilities, one for each action described. As illustrated in Figure 6, problem- and solution-specific input features are transformed into an embedding of length $D$ (we use $D = 64$), which is fed into an attention network (Vaswani et al., 2017) (we use an attention layer with 8 heads and 64 output units). The output of the attention network is concatenated with a sequence of recent actions and their effects (when $H > 0$). Lastly, the concatenated values are fed into a network of two fully connected layers, where the first layer uses 64 units and a Relu activation function and the second layer uses Softmax, producing $|A|$ action probabilities where $A$ is the set of actions.

### 2.1.4 REWARDS

We have experimented with a number of reward designs, two of which are producing satisfactory results as well as distinct patterns of operator sequences. The first reward function (denoted by RF1) focuses on the intermediate impact of the improvement operators. Specifically, the reward is +1 if the operator improves the current solution, -1 otherwise. The second reward function (denoted by RF2) is advantage-based. The total distance achieved for the problem instance during the first improvement iteration is taken as a baseline. For each subsequent iteration, all operators applied during this iteration received a reward equal to the difference between the distance achieved during the iteration and the baseline. We observed that an operator is often able to achieve a large distance reduction for a freshly perturbed solution, while it becomes harder and harder to do so in later improvement steps. In particular, the likelihood of distance reduction as well as the magnitude of such reduction, both decrease as the iteration proceeds. Therefore, it seemed unfair to give early improvement operators a larger reward. The observation suggested that operators used in the same improvement iteration should be rewarded equally and there would be no discounting (or equivalently, the discount factor $\gamma$ is 1).

To conclude the methodology section, we restart the improvement iteration until reaching a maximum number $T$ of rollout (either improvement or perturbation) steps. Following a common practice of encouraging exploration, we use $\epsilon$-greedy (Sutton & Barto, 2018) such that with a probability of 0.05 the RL controller will choose a random improvement action. Lastly, we also experimented with ensembling by training 6 different policies with $H = 1, 2, \ldots, 6$ (while keeping other components of the policy network identical). Ensembling facilitates learning of a diverse set of policies, as well as reducing wall-clock running time.

## 3 EXPERIMENTS AND ANALYSES

In this section, we present our experiment results. First we introduce a detailed setup of CVRP and hyper-parameters used. Then we compare our performance with prior neural network approaches, i.e., Nazari et al. (2018), Kool et al. (2019), Chen & Tian (2019), and a classic state-of-the-art heuristic algorithm. At last we provide detailed analysis of our framework[2].

**Setup and hyper-parameters**. We follow the same settings as previous works (Nazari et al., 2018; Kool et al., 2019; Chen & Tian, 2019) for CVRP. We consider three sub-problems with number of customers $N = 20, 50, 100$, respectively. The location $(x_i, y_i)$ of each customer, as well as of the depot, is uniformly sampled from unit square (specifically, $x_i$ and $y_i$ are uniformly distributed in the interval $[0, 1]$, respectively), and the traveling cost between two locations $c_{i,j}$ is simply the corresponding Euclidean distance. The demand $d_i$ of each customer is uniformly sampled from the discrete set $\{1, 2, \ldots, 9\}$. The capacity of a vehicle is 20, 30, 40 for $N = 20, 50, 100$, respectively. After $L = 6$ consecutive step of no improvement, we perturb the solution. To train the policy network, we use ADAM with a learning rate of 0.001. Unless otherwise stated, for a problem instance and a given policy we randomly initiate a feasible solution, and then iteratively update the solution $T = 40000$ times following the policy. In the following section we will discuss the performance of a policy with different rollout steps (recall that a rollout step is either an improvement step or a perturbation one). We choose the best one among all 40000 visited solutions as the final solution for a given problem instance. In the ensemble model we use different number of historical actions and effects to train a set of diverse policies (recall the policy network in Figure 6), and for any problem instance we choose the best solution produced by these policies. Unless otherwise stated, all reported metrics, such as the final traveling cost and the running time, are always computed as the average over 2000 random samples. Lastly, our method was implemented in Python, and the experiments were run computer nodes, each with a single Nvidia Tesla T4 GPU.

### 3.1 PERFORMANCE COMPARISON

In Table 1 we compare the performance of our algorithm with prior neural network approaches mentioned above, Google OR-tools (Google, 2019), and classic state-of-the-art heuristic algorithm LKH3 (Helsgaun, 2017). Our ensemble method chose the solution with minimum traveling cost among those produced by Policy $i$ (with $1 \leq i = h \leq 6$). As shown in Table 1, our algorithm outperforms the prior approaches in terms of average traveling cost. In particular, the average distance achieved by our algorithm is significantly shorter than prior neural network approaches. Thus, our algorithm is producing state-of-the-art results for CVRP, and is the first learning-based framework that outperforms the well-known classic heuristic algorithm.

Table 1: Comparison of our experiment results with those reported in the literature

|  | $N = 20$ Obj. | $N = 50$ Obj. | $N = 100$ Obj. |
| --- | --- | --- | --- |
| Google OR Tools | 6.43 | 11.31 | 17.16 |
| Nazari et al. (2018) | 6.40 | 11.15 | 16.96 |
| AM greedy (Kool et al., 2019) | 6.40 | 10.98 | 16.80 |
| AM sampling (Kool et al., 2019) | 6.25 | 10.62 | 16.23 |
| Chen & Tian (2019) | 6.16 | 10.51 | 16.10 |
| LKH (Helsgaun, 2017) | 6.14 | 10.38 | 15.65 |
| L2I | 6.12 | 10.35 | 15.57 |

### 3.2 ANALYSIS OF THE ENSEMBLE METHOD

Recall that we train 6 different policies with $H \in \{1, 2, \ldots, 6\}$. To illustrate the motivation of the ensemble method, we randomly pick 10 problem instances and show the traveling cost under different policies for these problem instances in Table 2. We can see that Policy 1 did the best for

---

[2]Code will be available at github.com/rlopt/l2i upon approval of the company.

the first problem instance, but it performed the worst for the second one. Furthermore, there is no clear winning policy, which is the reason we propose an ensemble method.

Table 2: Motivating examples: traveling cost under different policies for random problem instances

| | Problem instances | | | | | | | | | |
|---|---|---|---|---|---|---|---|---|---|---|
| | 1 | 2 | 3 | 4 | 5 | 6 | 7 | 8 | 9 | 10 |
| Policy 1 | 18.65 | 16.67 | 15.41 | 15.11 | 16.69 | 14.97 | 16.88 | 16.37 | 13.72 | 15.56 |
| Policy 2 | 18.86 | 16.61 | 15.39 | 15.10 | 16.55 | 14.75 | 16.73 | 16.39 | 13.77 | 15.56 |
| Policy 3 | 18.81 | 16.63 | 15.43 | 15.06 | 16.82 | 14.72 | 16.70 | 16.37 | 13.73 | 15.79 |
| Policy 4 | 18.90 | 16.60 | 15.37 | 15.04 | 16.66 | 14.93 | 16.83 | 16.32 | 13.70 | 15.62 |
| Policy 5 | 18.73 | 16.56 | 15.43 | 15.12 | 16.66 | 15.09 | 16.80 | 16.53 | 13.72 | 15.56 |
| Policy 6 | 18.92 | 16.60 | 15.43 | 15.09 | 16.73 | 14.65 | 16.59 | 16.55 | 13.73 | 15.56 |

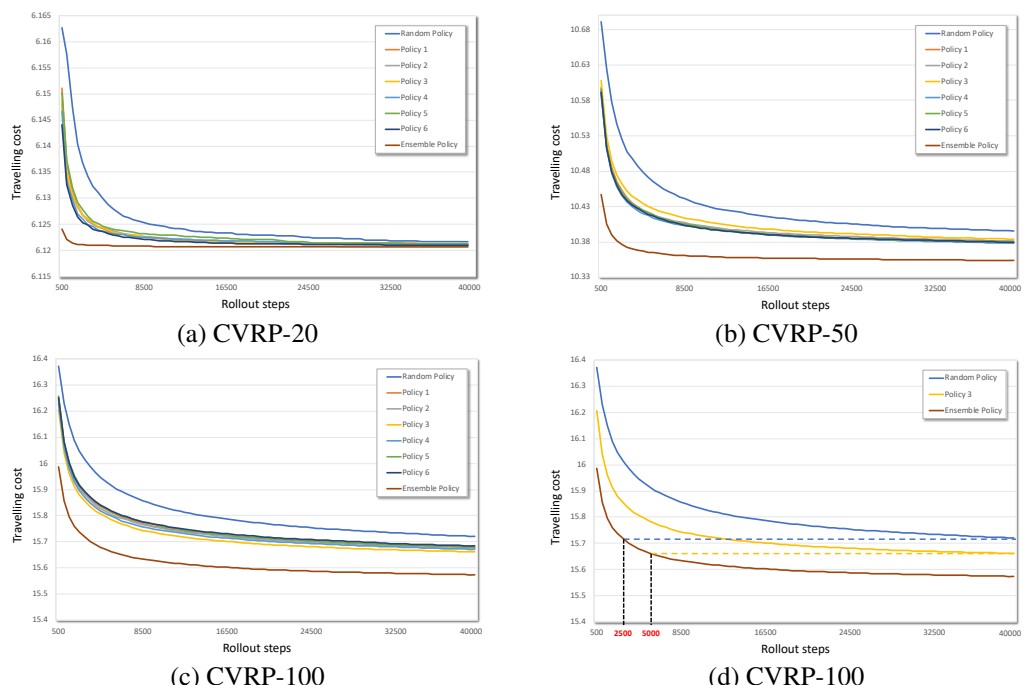

Figure 3: Average traveling cost of different policies with different rollout steps

In Figure 3(a), (b) and (c), we plot the average traveling cost over 2000 problem instances for CVRP with $N = 20, 50, 100$, respectively. It is worthwhile to point out that the same hyper-parameters are used for different $N$ values. The top blue line is for a random policy, the bottom red line is for our ensemble method, and lines in between are for Policy 1, 2, ..., 6. The plots show that, regardless of the number of rollout steps, our trained RL policies consistently outperform the random policy. Figure 4(c) shows that the gap between the random policy and the ensemble method first increases, peaks at about 5000 rollout steps, and then gets smaller and smaller. It is encouraging that we get the maximum gap fast, which will be helpful when computational time is limited. Lastly, in Figure 4(d) we show three policies, the random policy, the ensemble method, and the best RL policy (Policy 3), for CVRP-100. To match the performance of the random policy with 40000 rollout steps, the ensemble method would take 2500 rollout steps, while it takes 5000 rollout steps to match Policy 3 with 40000 rollout steps. The outperformance of the ensemble method becomes more obvious when $N = 20, 50$. These analysis shows that naively increasing rollout steps of a policy provides less marginal gains than ensembling a diverse set of policies.

Table 3: Improvement operators mostly used

| Class | Name | Details |
|---|---|---|
| Intra-route | 2-Opt | Remove two edges and reconnect their endpoints |
| | Relocate(1) | Move a customer in the route to a new location |
| Inter-route | Cross(2) | Exchange the tails of two routes |
| | Symmetric-exchange(2) | Exchange segments of length $m$ ( $m = 1$) between two routes |
| | Relocate(2) | Move a segment of length $m$ ( $m = 1$) from a route to another |

### 3.3 ANALYSIS OF OPERATOR USAGES

As we mentioned before, the RL model is able to differentiate more useful improvement operators from less useful ones for CVRP. In our experiments, we count the usage of different operators for different policies as training epochs grow. When the myopic reward function RF1 in Section 2.1.4 is used, our experimental results show that the policy converges to use a fixed subset of improvement operators (for detailed operators, see Table 3). This subset of operators are also preferred by all policies when we use RF2. However, the pattern of operator usages varies among the policies. For example, Figure 4(a) and (b) illustrate different patterns of operator usages for Policy 1 and Policy 2, respectively.

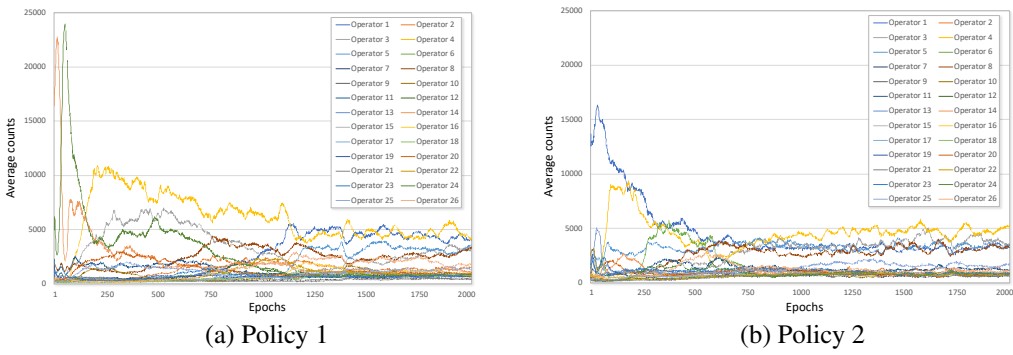

(a) Policy 1            (b) Policy 2

Figure 4: Pattern of operator usages as training epoch grows

### 3.4 ANALYSIS OF PERTURBATION MAGNITUDE

From our experimental study we also observed that the solution quality is sensitive to the magnitude of perturbation. To illustrate the impact of perturbation, in Figure 5 we plot the performance of two run configurations, one with Random-permute applied to all routes (called Random-permute-all), while the other with Random-permute applied to two routes only (called Random-permute-2). As seen from Figure 5, Random-permute-2 significantly outperformed Random-permute-all. Intuitively, when the magnitude of perturbation is too large, the resulting solution generally becomes much worse and it will take our algorithm a large number of improvement steps to remedy the deterioration.

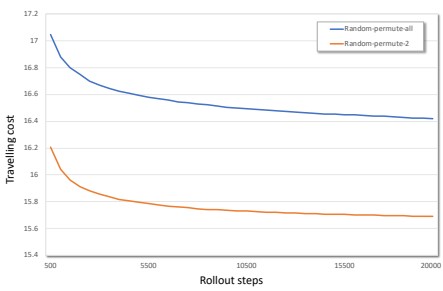

Figure 5: Impact of perturbation magnitude

## 4 CONCLUSION

In this paper we propose "Learn to Improve" for solving VRP, which starts with an initial solution and iteratively updates the solution with an improvement operator selected by an RL-based controller or with a perturbation operator chosen by a rule-based controller. We also propose an ensemble method that trains several RL policies and chooses the best solution produced by the policies. Our method achieved new state-of-the-art results for CVRP instances.

Our work provides a way of combining the strength of OR with learning capabilities of RL. For future work, we would like to apply the solution framework to solve other variants of the VRP, such as vehicle routing problems with time windows (VRPTW), as well as other combinatorial problems, such as maximum independent set problems and graph coloring problems. Furthermore, it is interesting to investigate whether allowing temporary constraint violations in our framework will help improve solution quality or not.

## 5 ACKNOWLEDGEMENT

We would like to thank our colleagues at Ant Financial, Wei Yan and Junping Zhao, for allocating GPU resources needed for the computational experiments.

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

## A  AN INTEGER PROGRAMMING FORMULATION OF THE CVRP

Formally, there is a depot and a set of $N$ customers in the CVRP. Each customer $i$, $i \in \{1, \ldots, N\}$, has a demand $d_i$ to be satisfied. A vehicle, which always starts at and ends at the depot, can serve a set of customers as long as the total customer demand does not exceed the capacity of the vehicle $C$. The traveling cost $c_{i,j}$ is the cost of a vehicle going from node $i$ to $j$, with $i, j \in V = \{0, 1, \ldots, N\}$ (where the depot is denoted by node 0 for convenience). The objective is to find a routing plan with minimal cost that serves all customers without violating vehicle capacity constraints. An integer programming formulation of the CVRP (Toth & Vigo, 2002) is given below.

$$\min_{x_{i,j}} \sum_{i \in V} \sum_{j \in V} c_{i,j} x_{i,j}$$

$$s.t. \sum_{i \in V} x_{i,j} = 1, \ \forall j \in V \setminus \{0\} \tag{1}$$

$$\sum_{j \in V} x_{i,j} = 1, \ \forall i \in V \setminus \{0\} \tag{2}$$

$$\sum_{i \in V} x_{i,0} = K, \tag{3}$$

$$\sum_{j \in V} x_{0,j} = K, \tag{4}$$

$$u_i - u_j + C x_{i,j} \leq C - d_j, \ \forall i,j \in V \setminus \{0\}, \ i \neq j, \ s.t. \ d_i + d_j \leq C \tag{5}$$

$$d_i \leq u_i \leq C, \ \forall i \in V \setminus \{0\}, \tag{6}$$

$$x_{i,j} \in \{0,1\}, \forall i,j \in V,$$

where $K$ is the number of vehicles available (w.l.o.g., it is assumed that $K = N$ for the CVRP we consider). Constraints (1) and (2) specify that each customer is visited exactly once, while constraints (3) and (4) specify the in and out degree of the depot, respectively. Constraints (5) and (6) impose the vehicle capacity requirements.

Table 4: State features

| Type | Name | Details |
|---|---|---|
| | $c_i$ | Demand of customer $i$ |
| | $C_i$ | Free capacity of the route containing customer $i$ |
| | $(x_i, y_i)$ | Location of customer $i$ |
| Problem- and solution-specific | $(x_{i-}, y_{i-})$ | Location of node visited before $i$ |
| | $(x_{i+}, y_{i+})$ | Location of node visited after $i$ |
| | $d_{i-,i}$ | Distance from $i^-$ to $i$ |
| | $d_{i,i+}$ | Distance from $i$ to $i^+$ |
| | $d_{i-,i+}$ | Distance from $i^-$ to $i^+$ |
| History-related | $a_{t-h}$ | Action taken $h$ steps before |
| | $e_{t-h}$ | Effect of $a_{t-h}$ |

## B  DETAILS OF STATES AND OPERATORS

We list the details of our state features in Table 4, and of operators in Table 5 and 6.

Table 5: Improvement operators

| Class | Name | Details |
|---|---|---|
| Intra-route | 2-Opt | Remove two edges and reconnect their endpoints |
| | Symmetric-exchange(1) | Exchange two customers in the route |
| | Relocate(1) | Move a customer in the route to a new location |
| Inter-route | Cross(2) | Exchange the tails of two routes |
| | Reverse-cross(2) | Reverse one of two routes and then exchange their tails |
| | Symmetric-exchange(2) | Exchange segments of length $m$ ($m = 1, 2, 3$) between two routes |
| | Asymmetric-exchange(2) | Exchange segments of length $m$ and $n$ ($m = 1, 2, 3$, $n = 1, 2, 3$, $m \neq n$) between two routes |
| | Relocate(2) | Move a segment of length $m$ ($m = 1, 2, 3$) from a route to another |
| | Cyclic-exchange(3) | Exchange cyclically one customer between three routes |

Table 6: Perturbation operators

| Class | Name | Details |
|---|---|---|
| Inter-route perturbation | Random-permute | Randomly destroy $m$ routes and re-construct routes by visiting affected customers in a random order |
| | Random-exchange(2) | Randomly exchange $m$ pairs of nearby customers between two routes |
| | Cyclic-exchange | Exchange cyclically customers between multiple routes |

## C  POLICY NETWORK

Figure 6 shows the structure of our policy network.

## D  RESULTS ON TSP

To quickly produce results for the TSP, we slightly modified our code for the CVRP by always using the first point in a TSP input as the depot in the CVRP formulation and enforcing that there is exactly one route in a solution. The capacity of any point is naturally assumed to be zero. Lastly, we implemented a simple perturbation operator by randomly permuting 20 points for TSP-50 and TSP-100 (10 points for TSP20). Figure 7 shows the trends of individual polices, as well as the ensemble one, for TSP-100. Again we observe that the ensemble method out-performed individual policies, while producing a total distance slightly above the optimal value of 7.76 as reported in the literature. The results for TSP-20 and TSP-50 are similar and thus omitted.

## E  SCALABILITY ANALYSIS

We also tested our method on larger CVRP instances, CVRP-200 and CVRP-1000[3], using the same experimental settings as CVRP-100 (e.g. the capacity of a vehicle is fixed at 50). The results are

---

[3]It is worthwhile to point out that CVRP-1000 results are averaged over 200, instead of 2000, instances, simply due to time limit.

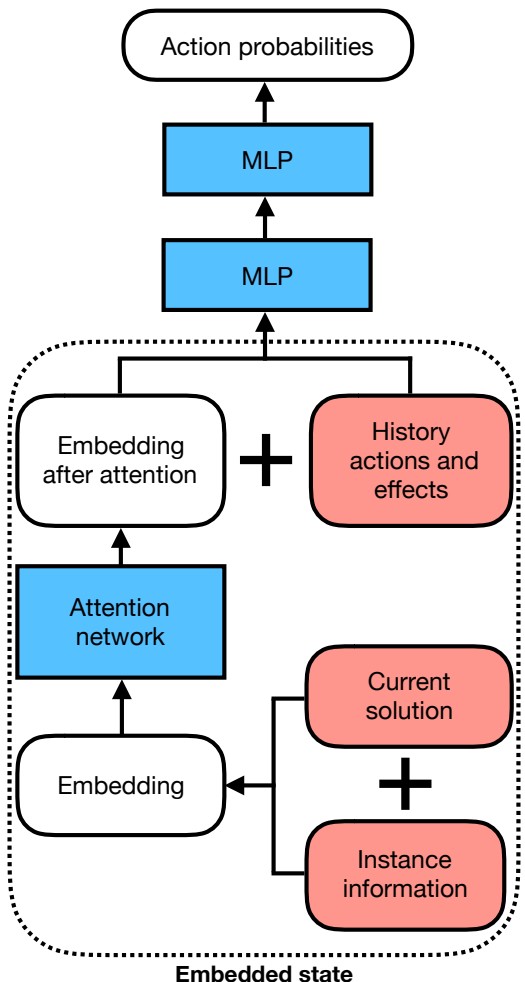

Figure 6: Policy network. The dash-line box is the state embedding part of policy network, which contains problem- and solution-specific input features, an attention network, and a sequence of historical actions and effects. The concatenated values are fed into a network of two fully connected layers, producing a vector of action probabilities.

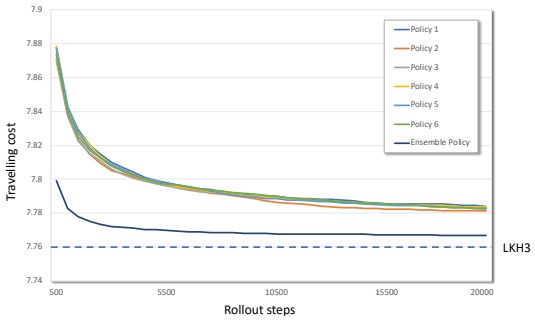

Figure 7: Results for TSP-100

given in Figure 8, which shows that our ensemble method scales well as the number of customers increases. In particular, the running time of our method increases less dramatically than LKH3.

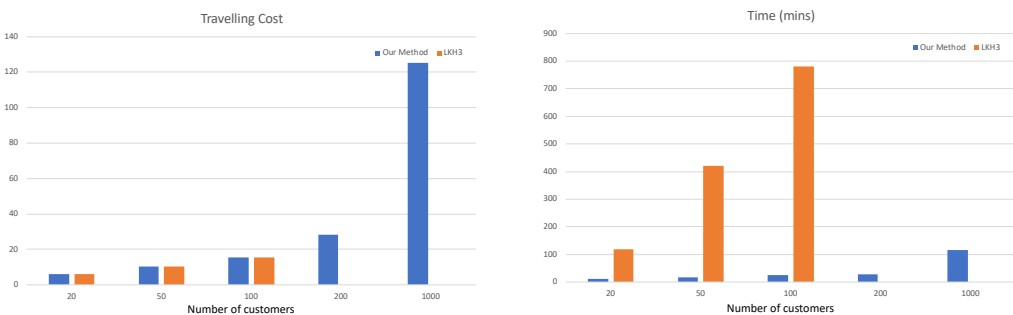

Figure 8: Traveling cost and average computation time as the number of customers increases

## F    SENSITIVITY ANALYSIS

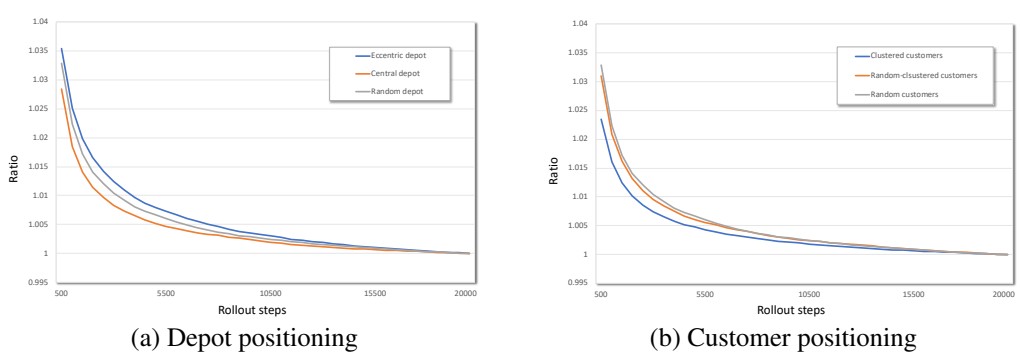

(a) Depot positioning                    (b) Customer positioning

Figure 9: CVRP-100 results under different data distributions by Policy 3

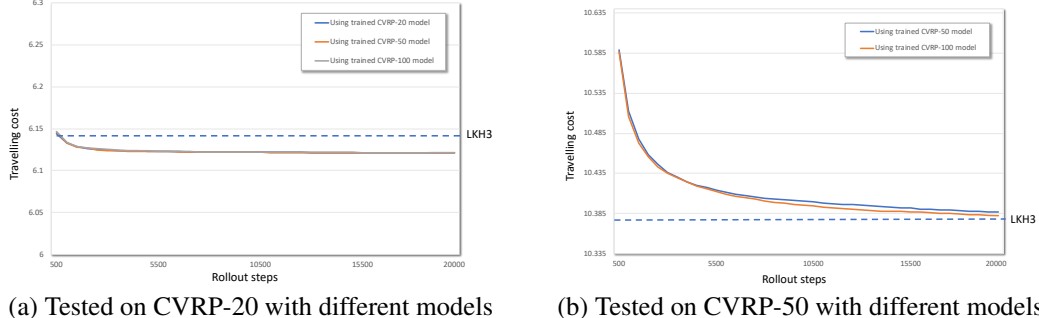

(a) Tested on CVRP-20 with different models          (b) Tested on CVRP-50 with different models

Figure 10: Generalization results using Policy 3

Following the same protocol of data generation as in Uchoa et al. (2017) (the authors of the paper were the creators and owners of the CVRPLib website[4]), we tested our method for four additional scenarios, namely, central depot positioning, eccentric depot positioning, clustered customer positioning, and random-clustered customer positioning. It is worthwhile to point out that the data distribution used in our initial paper corresponds to random depot and random customer positioning. Figure 9 (a) shows the impact of depot positioning (while using random customer positioning), and (b) shows the effect of customer positioning (while using random depot positioning). To make results from different data distributions comparable, we normalized the distance by the minimal distance achieved for each data distribution (denoted by "Ratio" as in the Figure), respectively. The plots show that our method works across different data distributions, and the decreasing trends of distance look similar.

---

[4]http://vrp.galgos.inf.puc-rio.br/index.php/en/

Furthermore, we also used the model trained for CVRP-100 to solve CVRP-50 and CVRP-20. Similarly, we also tested a trained CVRP-50 model for CVRP-20. Figure 10 shows that current implementation of our method can be applied to problems of different sizes.

