# OpenReview forum: "A Learning-based Iterative Method for Solving Vehicle Routing Problems"
_ICLR.cc/2020/Conference — Accept (Poster)_

### Official Review · AnonReviewer3 · 2019-10-09
**Official Blind Review #3**

**Rating:** 6

**Review:**

This paper proposes a framework combining RL and OR for solving the capacitated vehicle routing problem (CVRP). The main idea is to train an RL-based controller for choosing the OR operations necessary for improving & reinitializing the solution. Additionally, an ensemble method for diversifying the solution is proposed. The proposed method is shown to empirically outperform the existing OR method LKH3 for solving CVRP.

Overall, this paper provides a solid contribution by proposing a new learning-based algorithm that beats conventional solver, which is quite hard to achieve. The writing is clear and easy to read. The reference and related works look complete to my knowledge.

One weakness of the paper is the lack of experiments. Since the paper focuses on the CVRP problems, it would have been nice to compare with the LKH3 on larger graphs, e.g., the case when N > 100. One could also verify the scalability of the algorithm by examining larger-scale benchmarks and compare with heuristics faster than LKH3, e.g., see [1]. I think this is crucial for demonstrating the applicability of the algorithm to real-world problems.

[1] Uchoa et al., New benchmark instances for the Capacitated Vehicle Routing Problem, EJOR 2017

**Experience Assessment:**

I have read many papers in this area.

**Review Assessment: Checking Correctness Of Derivations And Theory:**

I did not assess the derivations or theory.

**Review Assessment: Checking Correctness Of Experiments:**

I assessed the sensibility of the experiments.

**Review Assessment: Thoroughness In Paper Reading:**

N/A

---

> ### Author Response · Authors · 2019-11-15
> **Response to Review #3**
>
> Thank you for your valuable comments. We conducted additional experiments to analyze the performance of our approach in solving larger-scale problems and in more diverse settings.
>
> First, we tested our solver on larger-sized problems to profile its scalability performance. The results have been appended to the paper (i.e., Appendix D), which shows that our ensemble method scales well as the number of customers increases. In particular, the running time of our method increases less dramatically than LKH3.
>
> Secondly, following the same protocol as the EJOR paper [1], we tested our method in 4 additional settings: 1) central depot positioning, 2) eccentric depot positioning, 3) clustered customer positioning, and 4)random-clustered customer positioning. Note that the data distribution used in our initial paper corresponds to random depot and random customer positioning. The results have been appended to the paper (i.e., Appendix E). Our method work fairly robustly across different data distributions.
>
> [1] Eduardo Uchoa, Diego Pecin, Artur Pessoa, Marcus Poggi, Thibaut Vidal, and Anand Subramanian. New benchmark instances for the Capacitated Vehicle Routing Problem, European  Journal of Operational Research, 257:845–858, 2017.

---

### Official Review · AnonReviewer2 · 2019-10-23
**Official Blind Review #2**

**Rating:** 6

**Review:**

EDIT: I have read the responses and increased my rating to Weak Accept.

The paper proposes an algorithm for the Capacitated Vehicle Routing problem that starts with a random solution and then iteratively improves it by using a learned policy to select an improvement operator to apply to the current solution. Once the solution stops improving for a number of steps, a perturbation is applied to the solution, and the iterations continue by applying improvement operators selected by the learned policy. The problem is posed as a sequential decision problem, and the policy is learned using reinforcement learning. Results on synthetic instances of size 20, 50, and 100 show that the approach is able to achieve better objective value than relevant baselines and faster running time than the LKH algorithm.

Pros:
- Better objective value than baselines and running time than LKH is an impressive result.

- Paper is clearly written and easy to follow. CVRP and other background information is sufficient to fully understand the paper.


Cons:
- The proposed approach has the benefit of using a handcrafted pool of improvement operators that other learning-based approaches don’t have. This makes the comparison to previous learning approaches unfair, and the observed improvements may simply be due to the extra information that the approach gets in the form of the operator pool.

- The evaluation should be expanded to larger instances and existing benchmarks such as CVRPLib (which contains larger instances as well) to understand scalability and generalization to different instance distributions. For example, would the same improvement operator pool suffice for a completely different distribution of instances?


Additional comments:
- Typically when using RL for learning a local search policy, the reward is defined as the change in objective after applying a local move such that the sum of rewards and the initial objective value gives the actual objective value and the policy will be learning to optimize the objective function. In this paper the reward definitions RF1 and RF2 don’t have this property, which suggests that the policy is not being trained to directly optimize the CVRP objective function. Is there an explanation for why it still works?

- It would be useful to explore how well the self-attention network architecture scales to larger problems (e.g., > 1000 customers), and whether using a graph neural network instead results in better or worse results.

- Typo: “significantly worse then”

- Typo: “their would be no discounting”


**Experience Assessment:**

I have published one or two papers in this area.

**Review Assessment: Checking Correctness Of Derivations And Theory:**

I assessed the sensibility of the derivations and theory.

**Review Assessment: Checking Correctness Of Experiments:**

I carefully checked the experiments.

**Review Assessment: Thoroughness In Paper Reading:**

I read the paper at least twice and used my best judgement in assessing the paper.

---

> ### Author Response · Authors · 2019-11-15
> **Response to Review #2 (part 2)**
>
> (2) "The evaluation should be expanded to larger instances and existing benchmarks such as CVRPLib (which contains larger instances as well) to understand scalability and generalization to different instance distributions. For example, would the same improvement operator pool suffice for a completely different distribution of instances?"
>
> We conducted additional experiments to analyze the performance of our approach in solving larger-scale problems and in more diverse settings.
>
> First, we tested our solver on larger-sized problems to profile its scalability performance. The results have been appended to the paper (i.e., Appendix D), which shows that our ensemble method scales well as the number of customers increases.
>
> Secondly, following the same protocol as the EJOR paper [6], we tested our method in 4 additional settings: 1) central depot positioning, 2) eccentric depot positioning, 3) clustered customer positioning, and 4)random-clustered customer positioning. Note that the data distribution used in our initial paper corresponds to random depot and random customer positioning. The results have been appended to the paper (i.e., Appendix E). Our method work fairly robustly across different data distributions.
>
> (3) "Reward Function"
>
> We observed that an improvement operator often leads to larger distance reduction on freshly perturbed solutions compared to later iteration steps. In particular, the likelihood of distance reduction as well as the magnitude of such reduction, both decrease as the iteration proceeds. Therefore, it seemed unfair to give early improvement operators a larger reward. RF1 (with +1 and -1 only) ignores the magnitude of the raw distance reduction in order to avoid biased rewarding. For RF2, our intuition is that operators used in the same improvement iteration should be rewarded equally, completely avoiding reward attribution. Regarding discount factors, in our experiments, we also tried different discount factors other than 1.0, but 1.0 seemed to give us better results. We've revised the text to make the description more clear.
>
> (4) "It would be useful to explore how well the self-attention network architecture scales to larger problems (e.g., > 1000 customers), and whether using a graph neural network instead results in better or worse results."
>
> With respect to scalability, please see our reply to (2). With respect to GNN, We agree that it could be promising to explore the idea in future work.
>
> [6] Eduardo Uchoa, Diego Pecin, Artur Pessoa, Marcus Poggi, Thibaut Vidal, and Anand Subramanian. New benchmark instances for the Capacitated Vehicle Routing Problem, European  Journal of Operational Research, 257:845–858, 2017.

---

> ### Author Response · Authors · 2019-11-15
> **Response to Review #2 (part 1)**
>
> Thank you for your valuable comments.
>
> (1) "The proposed approach has the benefit of using a handcrafted pool of improvement operators that other learning-based approaches don’t have. This makes the comparison to previous learning approaches unfair, and the observed improvements may simply be due to the extra information that the approach gets in the form of the operator pool."
>
> Existing approaches to solving combinatorial optimization problems such as CVRPs can be classified roughly into two types: 1) classical OR algorithms such as LKH3 that rely heavily on expert-designed heuristics and 2) emerging learning-based methods, which are purely data-driven. This paper was motivated by the considerable gap between these two types of methods: OR algorithms currently offer the state-of-the-art results but are in general slow and often difficult to scale (e.g., it takes LKH3 13+ hours to solve CVRP of size 100); in contrast, learning-based methods are often fast and scalable (once trained, they can finish solving within minutes or even seconds), but the quality of the solution is far behind OR algorithms (e.g., on CVRP the best learning-based solutions are significantly worse than LKH3's). The natural question is: is there a way to get the best of both worlds? In this paper, we show the answer is affirmative. In particular, by incorporating OR-guided operators into a learning-based procedure, our method is as fast as any other learning-based solvers, and at the same time, it produces solutions that are as good as OR methods (e.g., on CVRP, our method is able to beat LKH3 and establish the new state-of-the-art). From this perspective, our method is a hybrid of these two types. And hence yes, its usage of OR operators does give it an advantage when compared to other learning-based approaches. As a matter of fact, this is one of the key reasons that it could outperform them in problems like CVRP.
>
> Our experiments compared our method with three types of baselines: 1) classic OR algorithm LKH3, 2) pure learning-based approaches [3, 4] and 3) a recently-published algorithm [5] (published in upcoming NeurIPS'2019). We note that [5] is actually a hybrid algorithm which also uses a set of expert-designed OR rules. We show that our method is comparable in solving speed and scalability as [3,4,5] while beating them in solution quality by a large margin. Among all these approaches, hybrid methods (ours and [5]) are most promising as they are both fast and effective. For example, for CVRP-100, [5] achieved an average distance of 16.10, which is much better than 16.96 in [3] and 16.23 in [4]. The work [5] showed that a learning-based approach with OR guidance can improve previous learning methods. However, there is still a considerable gap as [5] is still far behind OR algorithms such as LKH3 (e.g.,  16.10 vs 15.65 on CVRP100). Learning-based approach to outperform LKH3 is challenging and non-trivial. Our method is the first hybrid method that's fast and scalable, and at the same time produces solutions that are on par or better than LKH3. The OR operators and the learning framework both played important roles in achieving the final result of 15.65, the new state-of-the-art on CVRP.
>
> [1] Oriol Vinyals, Meire Fortunato, and Navdeep Jaitly. Pointer networks. In Advances in Neural Information Processing Systems, pp. 2692–2700, 2015.
> [2] Keld Helsgaun. An extension of the Lin-Kernighan-Helsgaun TSP solver for constrained traveling salesman and vehicle routing problems. Technical report, Roskilde University, 2017.
> [3] Mohammadreza Nazari, Afshin Oroojlooy, Lawrence Snyder, and Martin Taka ́c. Reinforcement learning for solving the vehicle routing problem. In Advances in Neural Information Processing Systems, pp. 9839–9849, 2018.
> [4] Wouter Kool, Herke van Hoof, and Max Welling. Attention, learn to solve routing problems! In International Conference on Learning Representations, 2019.
> [5] Xinyun Chen and Yuandong Tian. Learning to perform local rewriting for combinatorial optimization. arXiv:1810.00337, to appear in NeurIPS 2019.

---

### Official Review · AnonReviewer1 · 2019-10-23
**Official Blind Review #1**

**Rating:** 6

**Review:**

Using a combination of RL-based solution as a warm start, this paper shows that by adding more improvement and perturbations, the quality of the solutions for capacitated VRP can be further improved. The numerical experiments show that with these extensions, the proposed mechanism is able to perform better than the SOTA LKH3 OR-based method in a shorter amount of time. I find the paper novel and interesting, but I have a few suggestions to further improve it:

1) Most of the mentioned operators are also valid for TSP. To provide a baseline for further research, I am expecting to see the performance of the proposed approach in TSP as well.
2) I hope that you make your code and saved models publicly available for future researchers since it might not be easy to replicate your numbers.
3) Maybe in future research, the ensemble idea can be tested for making the solution independent of problem size. This is an important possible extension.
4) That would be great if you could add a section for sensitivity analysis. How the model trained for VRP100 is working for say, VRP200, or VRP50?

Minor:
1) It worths mentioning how e-greedy that is primarily used in value-based methods is incorporated in policy gradient exploration.
2) Page 7, it should be "Figure 3(a), (b), ..."
3) Table 3 and Table 5 overlap. Maybe, it is better to merge them together.


**Experience Assessment:**

I have published one or two papers in this area.

**Review Assessment: Checking Correctness Of Derivations And Theory:**

I assessed the sensibility of the derivations and theory.

**Review Assessment: Checking Correctness Of Experiments:**

I carefully checked the experiments.

**Review Assessment: Thoroughness In Paper Reading:**

I read the paper thoroughly.

---

> ### Author Response · Authors · 2019-11-15
> **Response to Review #1**
>
> Thank you for your valuable comments.
>
> (1) We acknowledge that it's straightforward to apply our method to solve other routing problems such as TSPs, which in fact can be cast as a special class of CVRPs. Following the submission of this work, we tested our solver on TSPs. The code was slightly modified, for example only intra-route operators were allowed because TSPs are single-route problems. The results have been appended to the paper (i.e., Appendix C). Similar to the results on CVRPs, we observed that our ensemble method out-performed all the learning-based individual policies. On TSP-100, our solver was able to produce an average total distance that is slightly above the optimal value and that of LKH as reported in the literature.
>
> (2) Our code and all the saved models will be released with the publication of this paper.
>
> (3) Thank you for your suggestion. We will explore this idea in our future work.
>
> (4) Thanks for the suggestion. Sensitivity analysis results have been appended to the paper (i.e., Appendix D), where we tested models that were trained on problem of size X (e.g., CVRP-100) to solve problem of size Y (e.g., CVRP-50 and CVRP-20). The results show that our method can generalize to problems of different sizes fairly well.

---

> > ### Comment · AnonReviewer1 · 2019-11-15
> > **Concerns about 1**
> >
> > The reported optimal value is the literature by Bello et al., 2016 is 7.77, but you have obtained 7.76 which is, in fact, better than the optimal (surprisingly!). It can be due to the problems sampling errors, so please recompute the optimal for your sampled TSPs. Other than that, I am leaning toward acceptance of this paper.

---

> > > ### Author Response · Authors · 2019-11-15
> > > **Response to concerns about 1**
> > >
> > > Thank you for your comment.
> > >
> > > The value of 7.76, plotted as a horizontal line in Figure 7, is for LKH3 as reported in the literature [1]. As shown in the Figure, our results is slightly above the horizontal line.
> > >
> > > Indeed due to sample errors, there are slight differences for the optimal values for TSP-100, as well as, TSP-20 and TSP-50. Specifically, it is reported in [2] that the optimal values are
> > > TSP-20: 3.82,
> > > TSP-50: 5.68,
> > > TSP-100: 7.77,
> > > while in [1], they are
> > > TSP-20: 3.84,
> > > TSP-50: 5.70,
> > > TSP-100: 7.76,
> > > respectively.
> > >
> > > [1] Wouter Kool, Herke van Hoof, and Max Welling. Attention, learn to solve routing problems! In
> > > International Conference on Learning Representations, 2019.
> > > [2] Irwan Bello, Hieu Pham, Quoc V. Le, Mohammad Norouzi, and Samy Bengio. Neural combinatorial
> > > optimization with reinforcement learning. arXiv:1611.09940, 2016.

---

### Public Comment · ~Yansheng_Wang1 · 2023-05-06
**About Running Time of LKH-3**


Hi,

I really enjoy your work. But I still have some questions here. By my own experience of running LKH-3, it can always find the optimal solution to small problem size really fast (0.1s for 20 nodes, 2s for 100 nodes). But according to your results in Figure 8, it runs really slow (over 100 minutes for 20 nodes is very unusual) and fails to find the optimum, which is quite contrary to my common sense. Would you mind providing your implementation details of LKH-3 or give me some explanations?

Thanks a lot!

---

### Decision · Program_Chairs · 2019-12-19

**Decision:**

Accept (Poster)

**Comment:**

   The paper proposed the use of a combination of RL-based iterative improvement operator to refine the solution progressively for the capacitated vehicle routing problem. It has been shown to outperform both classical non-learning based and SOTA learning based methods. The idea is novel and the results are impressive, the presentation is clear. Also the authors addressed the concern of lacking justification on larger tasks by including an appendix of additional experiments.